Holobiont dysbiosis or acclimatation? Shift in the microbial taxonomic diversity and functional composition of a cosmopolitan sponge subjected to chronic pollution in a Patagonian bay

Gastaldi Marianela 1 2
http://orcid.org/0000-0002-7061-9613 Pankey M. Sabrina 3
Svendsen Guillermo 1 4 guillermosvendsen@gmail.com
Medina Alonso 1
Firstater Fausto 1 2
Narvarte Maite 1 2
Lozada Mariana 5
Lesser Michael 3
1 Escuela Superior de Ciencias Marinas, Universidad Nacional del Comahue , San Antonio Oeste, Río Negro , Argentina
2 Laboratorio de Biodiversidad y Servicios Ecosistémicos, CIMAS-CONICET , San Antonio Oeste, Río Negro , Argentina
3 Department of Molecular, Cellular and Biomedical Sciences and School of Marine Science and Ocean Engineering, University of New Hampshire , Durham, New England , United States
4 Laboratorio de Modelado Ecológico y Pesquero, CIMAS-CONICET , San Antonio Oeste, Río Negro , Argentina
5 Laboratorio de Microbiología Ambiental, IBIOMAR-CONICET , Puerto Madryn, Chubut , Argentina
Kormas Konstantinos
Electronic publication date: 2024 Aug 21
Publication date: 2024
Volume: 12
Electronic Location ID: e17707
Received 2024 Feb 5; Accepted 2024 Jun 18
Copyright: © 2024 Gastaldi et al.
Copyright year: 2024
Copyright holder: Gastaldi et al.
License: This is an open access article distributed under the terms of the Creative Commons Attribution License, which permits unrestricted use, distribution, reproduction and adaptation in any medium and for any purpose provided that it is properly attributed. For attribution, the original author(s), title, publication source (PeerJ) and either DOI or URL of the article must be cited.
License URL: https://creativecommons.org/licenses/by/4.0/

Keywords: Hymeniacidon perlevis, Anthropogenic impact, Host-microbe association, Holobiont, Water chronic pollution

Funding: CONICET Postdoctoral Scholarship Company of Biologists with a Travelling Fellowship JEBTF-170808 National Science Foundation, Dimensions of Biodiversity OCE-1638296 Marianela Gastaldi was supported by a CONICET postdoctoral scholarship and by the Company of Biologists with a Travelling Fellowship (Grant JEBTF-170808). Sabrina Pankey and Michael Lesser were supported by a National Science Foundation, Dimensions of Biodiversity grant (OCE-1638296). The funders had no role in study design, data collection and analysis, decision to publish, or preparation of the manuscript.

==============================
Dysbiosis and acclimatization are two starkly opposing outcomes of altered holobiont associations in response to environmental pollution. This study assesses whether shifts in microbial taxonomic composition and functional profiles of the cosmopolitan sponge Hymeniacidon perlevis indicate dysbiotic or acclimatized responses to water pollution. To do so, sponge and water samples were collected in a semi-enclosed environment (San Antonio Bay, Patagonia, Argentina) from variably polluted sites (i.e., eutrophication, heavy metal contamination). We found significant differences in the microbiome of H. perlevis with respect to the pollution history of the sites. Several indicators suggested that acclimatization, rather than dysbiosis, explained the microbiome response to higher pollution: 1) the distinction of the sponge microbiome from the water microbiome; 2) low similarity between the sponge and water microbiomes at the most polluted site; 3) the change in microbiome composition between sponges from the different sites; 4) a high similarity in the microbiome among sponge individuals within sites; 5) a similar ratio of common sponge microbes to opportunistic microbes between sponges at the most and least polluted sites; and 6) a distinctive functional profile of the sponge microbiome at the most polluted site. This profile indicated a more expansive metabolic repertoire, including the degradation of pollutants and the biosynthesis of secondary metabolites, suggesting a relevant role of these microbial communities in the adaptation of the holobiont to organic pollution. Our results shed light on the rearrangement of the H. perlevis microbiome that could allow it to successfully colonize sites with high anthropogenic impact while resisting dysbiosis.

Introduction

Coastal zones are valuable ecosystems, often under intense anthropogenic pressure (He & Silliman, 2019). One of the main causes of coastal ecosystem degradation is pollution directly caused by anthropogenic activities (Häder et al., 2020). Specifically, cultural eutrophication and heavy metal contamination of coastal waters lead to deleterious effects on marine organisms, affecting food webs, water quality, and chemistry (Rabalais et al., 2009; Häder et al., 2020; Malone & Newton, 2020). This scenario seems difficult to reverse when an increase of human populations along coastal ecosystems worldwide is projected to be from 50% to 120% for the period 2030–2060 (Neumann et al., 2015). However, some organisms can not only withstand these increasingly adverse conditions but benefit the greater ecosystem by removing pollutants. This is the case for sponges, conspicuous and functionally important members of benthic communities, capable of filtering enormous volumes of water and serving as valuable bioindicators (Bell et al., 2013; Girard et al., 2021). These sessile, filter-feeding animals are among the most ancient organisms on Earth (Love et al., 2009; Yin et al., 2015), and have successfully colonized a wide range of habitats around the globe including highly eutrophic environments (Manconi & Pronzato, 2008; Van Soest et al., 2012; Pomponi et al., 2019). Their evolutionary success is related to their associated microorganisms (microbiome), which perform various metabolic, physiologic, and immune functions that benefit the sponge host, contributing to its homeostasis (Hentschel et al., 2012; Slaby et al., 2019; Lesser et al., 2022; Pankey et al., 2022). Given that this intimate host-microbe association functions as a cohesive, co-diversifying evolutionary unit, sponges are considered to constitute “holobionts” together with their microbiome (sensu Pita et al., 2018; Li, 2019; Pankey et al., 2022). According to the abundance, density, and taxonomic composition of these associated microbes, sponges are classified in two distinct groups: low microbial abundance (LMA), which is the ancestral state among sponges; and high microbial abundance (HMA), which harbor densities of microbes 2–4 orders of magnitude higher than LMA sponges, and present increased metabolic dependence and chemical defense (Pankey et al., 2022).

Health status of sponge holobionts against anthropogenic stress factors

The general paradigm posits that sponge-microbes association remains stable (i.e., similar microbial composition) under environmental disturbance (Taylor et al., 2007; Simister et al., 2012b; White et al., 2012; Pita et al., 2013; Cárdenas et al., 2014; Luter, Gibb & Webster, 2014), resulting in the resistance and resilience of the sponge holobiont. Different anthropogenic drivers, however, can alter the degree of this association, resulting in changes of the holobiont’s health status (reviewed in Slaby et al., 2019; Yang, Zhang & Franco, 2019). Altered associations can lead to two opposing outcomes: dysbiosis or acclimatization. Dysbiosis is characterized by the loss of homeostatic microbial functions (i.e., metabolic and defensive functions; reviewed in Hentschel et al., 2012; Webster & Thomas, 2016), increasing risk of pathogen infection and death (Egan & Gardiner, 2016; Pita et al., 2018). On the other hand, acclimatization promotes holobiont survival under disturbance conditions by introducing or reconfiguring beneficial microbial functions (Marangon et al., 2021). Acclimatization precedes adaptation, in which holobiont fitness increases due to heritable microbiome changes, or host/symbiont evolution (Pita et al., 2018; Marangon et al., 2021). In both dysbiotic and acclimatized outcomes, microbial assemblages change in terms of richness and composition (i.e., Turque et al., 2010; Lesser et al., 2016; Vargas, Leiva & Wörheide, 2021). However, the direction and magnitude of these changes can vary greatly depending on the nature of the environmental disturbance and the identity of the sponge host, making it difficult to find clear patterns to discern between these two outcomes.

Evidence suggests that sponge-microbe associations are often resilient to pollution disturbance (e.g., nutrients, heavy metals). Two microcosm studies, where the sponges Cymbastela stipitata and Rhopaloeides odorabile were exposed to short-term elevated nutrient concentrations, showed no shift in microbial composition compared to control treatments (Simister et al., 2012b; Luter, Gibb & Webster, 2014). Field studies examining the effects of pollution on the microbiomes of sponges Amphimedon paraviridis and Crambe crambe showed a similar trend, where microbial diversity measures did not differ between sites with contrasting pollution impact (Gantt, López-Legentil & Erwin, 2017; Turon et al., 2019). On the other hand, the microbiome of Hymeniacidon heliophila inhabiting the polluted waters of Guanabara Bay in Brazil, showed an increase in archaeal richness, and differences in microbial taxonomic composition compared to sponges inhabiting less polluted offshore waters at Cagarras Islands (Turque et al., 2010). The discrepancies among these findings highlight the need for comprehensive field studies on sponge microbial assemblage indicators exposed to different pollution conditions.

Anthropogenic influence in the San Antonio Bay

The city of San Antonio Oeste (SAO) is located on the west coast of San Antonio Bay (SAB), with a population over 35,000 people (Argentina 2022 census, www.indec.gob.ar) (Fig. 1). The city is settled on unconsolidated sandy sediments, and lacks efficient septic systems, resulting in domestic wastewater entering the groundwater and flowing through the aquifer to the nearby channel (SAO channel) since the foundation of the city (~120 years). This freshwater input is nitrogen-rich and directly affects the structure and functioning of the aquatic communities living in the channel (Martinetto et al., 2010, 2011; Teichberg et al., 2010; Fricke et al., 2016; Becherucci et al., 2019; Marello Buch et al., 2024). This channel also receives the direct discharge of untreated effluents from fish processing facilities located in the city (Giaccardi & Reyes, 2014). These discharges do not reach adjacent channels, which have significantly lower nutrient concentrations (42%, 88% and 92% lower concentration of nitrate, ammonium, and phosphate; Martinetto et al., 2010; Fricke et al., 2016; Marello Buch et al., 2024). Moreover, mining wastes of an old mine located at the upland west border of the SAO channel have been wind and water dispersed to the channel for the last six decades (Häder et al., 2020). As a result, high concentrations of lead, zinc, copper and iron have been detected in mollusks and saltmarsh plants along the SAO channel (Vázquez et al., 2007; Idaszkin et al., 2015), as well as in the human population of San Antonio Oeste (https://multisectorialplomo.org/). The concentrations of these heavy metals decrease along this channel from the innermost sites to the mouth of SAB (Idaszkin et al., 2015; Marinho et al., 2017).

Figure 1 Study site and pollution sources.

(A) San Antonio Bay is located in northern Patagonia, Argentina. (B) Samples were collected at three sites in the west coast of the bay. (C) Detail of the sampling sites with different historical levels of pollution (low, medium, and high, see legends in (B)). Sampling sites consisted of an area of approximately 0.25 ha. In (C) the red rectangle indicates the location of the mining wastes, the green rectangle indicates the fishery processing facility, and the blue arrows show the direction of heavy metals and nutrients input from the city septic system to the main channel. The description of pollution levels can be found in Table S1. The map composition was created in QGIS using Google Satellite imagery. The attributes of each Google Satellite image is: (A) Images © 2024 NASA, Terra Metrics, Map data © 2024; (B) Images © 202, Terra Metrics, Map data © 2024; (C) Images © 2024 Airbus, CNES/Airbus, Maxar Technologies, Mapdata © 2024.

Here, we studied the sponge Hymeniacidon perlevis (Montagu, 1814–1818) to understand the effects of pollution on the structure and function of its microbiome and to assess the stability of the association between the microbiome and the sponge host. Hymeniacidon perlevis is a cosmopolitan sponge well adapted to a wide range of environmental conditions and a variety of heavily impacted habitats around the world such as the Venice Lagoon (Italy), and many semi-enclosed and enclosed basins with urban, industrial and fish farms waste discharges (Stone, 1970; Corriero et al., 2007; Xue, Zhang & Zhang, 2009; Longo et al., 2022). This sponge was recently identified in SAB, Southwest Atlantic coast, northern Patagonia Argentina, where it is the most abundant sponge (Gastaldi et al., 2016). Previous studies have shown visually healthy H. perlevis individuals inhabiting highly polluted sites inside the bay (Gastaldi et al., 2017, 2020), where high concentrations of nutrients, particulate matter, and heavy metals have been reported (Martinetto et al., 2010; Teichberg et al., 2010; Fricke et al., 2016; Saad et al., 2019; Häder et al., 2020). Taken together, these findings suggest this sponge to be acclimatized and thrive successfully under polluted conditions. However, there is no information on the role of the microbiome in the ability of this holobiont to inhabit such impacted environments.

This study finds that H. perlevis exposed to chronic water pollution exhibit shifts in microbial taxonomic composition and functional profiles consistent with acclimatization rather than dysbiosis. We hypothesize that, in the SAB, the successful colonization of heavily impacted sites by H. perlevis is related to the reshaping of its microbiome, which facilitates the successful holobiont acclimatization in this impacted environment.

Materials and Methods

Study area

The sampling was performed in December 2017 in San Antonio Bay (SAB; Northern Patagonia, Argentina; Fig. 1). This bay is characterized by strong westerly winds, with a seasonal precipitation pattern (0.1–7 mm month−1), and high thermal amplitude (Cabrera, 1976; Paruelo et al., 1998). Mean annual atmospheric temperature is 15 °C, with extreme records in July and February (winter: −8.1 °C; summer: 41.4 °C); the mean annual humidity is 57 ± 13%, and wind speed can exceed 60 km h−1 (Lucas et al., 2005; Genchi et al., 2010; Fricke et al., 2016). The bay is dominated by tidal currents, with <0.5 m s−1 in the inner bay, increasing near the mouth to 2 m s−1 (Schnack et al., 1996). Due to its low average precipitation (250 mm yr−1), lack of freshwater input via precipitation or from watercourses and high evaporation, SAB is generally hypersaline.

This study was carried out in three sites representative of contrasting pollution states. From the city channel (SAO channel) we selected a site in the inner region of the channel, which represents the high-pollution condition (Fig. 1, Table S1). From the adjacent channel (CTL channel) we selected a site (0.63 km away from the highly polluted site) representing a control condition, with low-pollution impact. Finally, Punta Verde (PV), located in the confluence of the previous two channels (3.35 and 2.25 km from the reference sites with high and low pollution), was selected as a site representative of medium-pollution condition (Fig. 1, Table S1) (Saad et al., 2019; Marello Buch et al., 2024).

Sample collection

Seawater and H. perlevis samples were collected from the shallow subtidal (25 to 50 cm depth) during low tide from an area of approximately 0.25 ha at each site (Figs. 2A–2C, n = 5 sponges, and water samples per site). Sponges were photographed in situ and sampled into sterile WhirlPak bags (Nasco, Saugerties, NY, USA) underwater. In the laboratory, specimens were visually inspected for health before sampling following the consistency, surface, color and smell characteristics detailed in Gastaldi et al. (2018). Seawater was collected into acid-washed bottles and 25–50 mL were concentrated onto 0.2 μm polycarbonate filters. Tissue samples and filters were preserved in Seutin’s buffer (Seutin, White & Boag, 1991) and stored at −20 °C for genetic analyses.

Figure 2 Hymeniacidon perlevis from the sampling sites.

Healthy individuals collected at sites with different levels of pollution: (A) low, (B) medium, and (C) high. The description of pollution levels can be found in Table S1. Scale bar in (A) is the same for (B and C).

Tissue sampling and DNA extraction, amplification, and sequencing

A piece of each sponge sample (~200 mg), comprising the full thickness of the sponge body, was dissected for genomic analyses. Total genomic DNA was extracted from each sample using DNeasy PowerSoil Kit (Qiagen, Hilden, Germany) following the manufacturer’s instructions with modifications to cell lysis as described in Pankey et al. (2022). We carried out phylogenetic analyses to assess population differences in the genotypes of sponges from different sites that might be related with differences in their microbial assemblages. To do so, we amplified sponge nuclear and mitochondrial (ITS2, 18S, COI) genes (see Article S1 for ITS2, 18S, COI amplification and sequencing details).

Microbial DNA amplification and sequencing

Microbial 16S rRNA gene was amplified by PCR from the total DNA extractions of sponge tissue and water samples, using a universal bacterial/archaeal primer set (hypervariable region V3–V4), consisting of the forward primer 515F (5′-GTGYCAGCMGCCGCGGTAA; Parada, Needham & Fuhrman, 2016) and the reverse primer 806R (5′-GGACTACN-VGGGTWTCTAAT; Apprill et al., 2015). Fluidigm linker sequences CS1 (5′-ACACTGACGACATGGTTC-TACA) and CS2 (5′-TACGGTAGCAGAGACTTGGTCT) were added to the 5′ end of both forward and reverse primers to facilitate Illumina MiniSeq. See Article S1 for amplification and pipeline details.

Analytical approach to distinguish between holobiont outcomes

We analyzed a set of indicators of the microbiome α- and β-diversity, and functions which, taken together, allows us to assess the effect of pollution on the structure and functioning of the H. perlevis microbiome. Whereas no change in these indicators is a sign of the microbiome resistance, their variation allows us to discern between different states of the association -acclimatization and dysbiosis- of the sponge and its microbiome.

The taxonomic α-diversity indicators are: ASV richness (Turque et al., 2010; Vargas, Leiva & Wörheide, 2021); and ASV evenness. Some hypotheses can be generated based on studies reporting changes in the dominance of different microbes in holobionts experiencing early signs of disease. Blanquer et al. (2016) showed a decrease in microbial dominance, which translates into an increase in the evenness in holobionts with early signs of disease. Thus, we predict a lower evenness to be a sign of acclimatization (Fig. 3A), while a higher evenness is a sign of dysbiosis (Fig. 3B).

Figure 3 Scheme of microbial shifts under different scenarios.

(A) Sponge microbial evenness under acclimatization, (B) under dysbiosis, (C) microbial compositional similarity under acclimatization, and (D) under dysbiosis. In panels (A and B), dots with different colors represent sponge microbes. (C and D) Represent the similarity of the samples in a 2D ordination space (adapted from Zaneveld, McMinds & Vega Thurber, 2017). Orange and blue sponges represent sponge samples from low- and high-pollution sites, respectively; stars represent water samples; yellow arrows indicate similarity between microbiomes from sponge and water samples; blue arrows indicate inter-site microbial similarity; and red arrows indicate intra-site microbial similarity.

Taxonomic β-diversity: acclimatization of holobionts is associated with low similarity between microbial compositions of the host and environment (Vargas, Leiva & Wörheide, 2021). Conversely, dysbiosis results in increased similarity between the microbial compositions of the host and the environment (Pita et al., 2018). Moreover, a decrease in microbiome similarity among sponges from the same site/treatment compared to the similarity among sponges from different sites/treatments is a sign of dysbiosis (Lesser et al., 2016; Pineda et al., 2016; Turon et al., 2019). Given this prior knowledge, we predict that acclimatized sponges will display i) a distinct sponge microbiome (with low similarity to water microbiome); ii) low similarity among the microbial compositions of sponges from different sites (low inter-site similarity); iii) a high similarity in the microbial composition among individuals from the same site (high intra-site similarity) (Fig. 3C). Conversely, sponges experiencing dysbiosis will display: i) high compositional similarity to water microbiomes; ii) similarity of their microbiomes increase towards the more polluted site (inter-site); iii) low microbial similarity among sponge individuals from the same site (intra-site) (Fig. 3D).

Microbes that determine changes in composition: We studied the contribution of specific taxa to the observed differences between sources and among sites with edgeR (McCarthy, Chen & Smyth, 2012). edgeR implements statistical methods based on the negative binomial distribution as a model for count variability. Those ASVs with FDR-values less than 0.05 and Log2 fold-change values >6 or <−6 (which returned those ASVs that were exclusive at a site and excluded the differentially expressed ASVs present at both sites) were considered.

Ratio of common (among sponges) to opportunistic microbes: A higher ratio is expected to occur in healthy hosts compared to dysbiotic hosts, in which assemblages are enriched with microbes closely related to opportunistic or free-living microbes, and/or depleted of microbes commonly associated with sponge species (‘endemic’) (Webster, Cobb & Negri, 2008; Fan et al., 2013; Botté et al., 2023).

Microbial functions: Acclimatized holobionts can experience an expansion in their metabolic capabilities, including functions that detoxify the sponge tissue and prevent disease via inhibition of the onset of pathogens, which increase holobiont success to the new conditions (Turque et al., 2010; Marangon et al., 2021). In contrast, an enrichment in functions related to scavenging lifestyle, such as cell motility, chemotaxis, and rapid growth characterize dysbiotic holobionts (Fan et al., 2013; Posadas et al., 2022).

Raw counts data were used for index calculations and analyses. Samples were not rarefied either, since rarefaction and proportions transformation suffer from a failure to address overdispersion among biological replicates, with rarefied counts also suffering from a loss of power, and proportions failing to account for heteroscedasticity (McMurdie & Holmes, 2014). Plots were constructed with ggplot2() (Wickham, 2016), and all the analyses were performed in RStudio (RStudio Team, 2021). See Article S1 for statistical analysis details.

Nucleotide sequence accession numbers

The sequences determined in this study have been submitted to GenBank under the accession numbers MZ292042, MZ297336–MZ297354 (for COI sequences); MZ298268–MZ298287 (for 18S sequences); and MZ435928–MZ435947 (for ITS2 sequences). Microbiome 16S rRNA libraries have been deposited in NCBI Short Read Archive under the BioProject PRJNA734169.

Results

Sponge genotype

Hymeniacidon perlevis from different sites at the SAB showed no intraspecific genetic variation (Fig. S1). The recovered COI, 18S, and ITS2 sequences support monophyly between H. perlevis samples from SAB, Ireland, Caribbean, USA, China, and Korea (Fig. S1), clustered separately from the sequences of H. caerulea from Panama and H. flavia from Korea and Japan. Table S2 shows the accession numbers of COI, 18S, and ITS2 sequences from this study and other studies included in the analyses.

Sponge and water microbial assemblages

Fifteen sponge samples were sequenced (n = 5 for each site). In the case of the water samples, three of the five samples from the high-pollution site were sequenced for a total of 13 sequenced samples (n = 5 for the low- and medium-pollution sites). The sponge and water 16S samples produced 26,645 ± 3,735 sequences (mean ± SD) with a Good’s coverage of 0.9995 ± 0.0004 (with min-max of 0.998374–0.999902) (Table 1). A total of 4,775 unique ASVs were identified in the samples and assigned to 54 known phyla of bacteria and seven archaea. Unidentified ASVs represented 3% and 7% for Bacteria and Archaea, respectively (154 and 64 ASVs, respectively). Sponge and water samples presented a total of 1,765 (296.8 ± 92.9 ASVs per sample) and 3,529 (549.4 ± 166) ASVs, respectively. The water and sponge samples consisted mainly of the phylum Pseudomonadota (60% and 78%, respectively), followed by Bacteroidota (22% and 13%, respectively). In the water samples, the phyla Nanoarchaeota, Verrucomicrobiota, Euryarchaeota, Actinomycetota, Chloroflexota, Planctomycetota, Epsilonbacteraeota, Patescibacteria, Kiritimatiellaeota, Nitrososphaerota, Cyanobacteriota and Firmicutes contributed between 2.4% to 0.5%, and 37 less abundant phyla contributed less than 0.5% (Fig. 4, Table S3). In the case of sponges, the phyla Cyanobacteriota, Planctomycetota, Verrucomicrobiota, Actinomycetota, Nitrososphaerota, Chloroflexota, Spirochaetes and Acidobacteriota contributed between 2% and 0.5%, and 28 less abundant phyla contributed less than 0.4% to the sponge microbial abundance (Fig. 4, Table S3).

Table 1 α-diversity metrics (mean ± sd) of the microbiome of Hymeniacidon perlevis and water samples.

Low, medium, high represents sites with different levels of pollution. A further description of the sites can be found in Table S1.

Source	Site	Observed richness	Simpsom evenness	Good’s coverage	
Sponge	Low	246 ± 65	0.05 ± 0.02	0.99981 ± 8E−5	
Sponge	Medium	256 ± 44	0.10 ± 0.02	0.99982 ± 6E−5	
Sponge	High	388 ± 91	0.09 ± 0.01	0.99967 ± 1.4E−4	
Water	Low	726 ± 97	0.05 ± 0.02	0.99887 ± 4.1E−4	
Water	Medium	421 ± 94	0.06 ± 0.03	0.9995 ± 1.8E−4	
Water	High	470 ± 19	0.02 ± 0.001	0.99931 ± 9E−5	

Figure 4 Relative abundance of Hymeniacidon perlevis microbes.

(A) Bacteria and (B) Archaea phyla from sponge and water samples. Low, medium, and high refers to sites with different levels of pollution. A further description of the sites can be found in Table S1.

Holobiont health state indicators

Taxonomic α-diversity

ASV richness differed between sample sources, sites, and the interaction term (ANOVA, source*site: F2,22 = 17, P < 0.001; Table 1), which means that the difference in richness between sponge and water microbiomes did not remain constant among sites. That is, the difference in richness decreased from the low-pollution site until it became similar at the high-pollution site (Tukey test results; Fig. 5A). ASV evenness also differed among sample sources, sites, and the interaction term (ANOVA, source*site: F2,22 = 5.4, P < 0.05; Table 1). The water samples displayed significantly lower evenness than the sponge samples. The difference in evenness increased between sponge and water samples from the low- to high-pollution sites (Tukey test results; Fig. 5B).

Figure 5 α-diversity of microbial assemblages of Hymeniacidon perlevis and water samples.

(A) Observed richness and (B) Simpson evenness. Low, medium, and high represent sites with different levels of pollution (see Table S1). Lower-case letters indicate significant differences between sources and sites (Tukey’s post hoc test). Black dots and lines represent means and standard deviation, respectively.

Taxonomic β-diversity

Differences in the assemblage composition at the ASV level between sponge and water samples were contingent on the site (PERMANOVA, source*site: F4,27 = 1.9, P < 0.001; Tables 2, 3). Individual PERMANOVA showed sponge and water microbiome compositions to be different at each site (Tables 2, 3), which was evidenced in the nMDS as separate groups (Fig. S2). The dissimilarity between sponge and water microbiomes was the highest at low- and high-pollution sites, whereas it was the lowest between sponge and water samples from the medium-pollution site (ANOVA, F2,62 = 21.5, P < 0.001; Tukey test results; Fig. S3). The composition of the sponge microbiomes differed among sites (PERMANOVA, F2,14 = 2.3, P < 0.001), with the lowest dissimilarity found between low- and medium-polluted sites, while the highest dissimilarity was found between low- and high-polluted sites (multilevel pairwise comparison test; all P-adjusted values <0.05; Table 2). The intra-site dissimilarity in the microbial composition among sponge individuals was similar among the sites (BETADISPER, P = 0.27; Fig. S2).

Table 2 Jaccard dissimilarity distances (mean ± sd) of the microbiome from different sources (sponge, water) and sites.

Low, medium, and high represent sites with different levels of pollution. A further description of the sites can be found in Table S1.

	Jaccard distance	
Sponge-water dissimilarity		
Low	0.89 ± 0.02	
Medium	0.81 ± 0.05	
High	0.88 ± 0.03	
Inter-sites sponge dissimilarity		
Low-medium	0.73 ± 0.03	
Low-high	0.81 ± 0.04	
Medium-high	0.77 ± 0.05	
Intra-site sponge dissimilarity		
Low	0.72 ± 0.04	
Medium	0.64 ± 0.04	
High	0.70 ± 0.05	

Table 3 PERMANOVA table results based on Jaccard dissimilarity distances.

Source refers to sponge and water samples. Sites refers to sampling sites with different levels of pollution (low, medium, high). A further description of the sites can be found in Table S1.

	DF	F	P (perm)	
Two-ways analysis				
Source	1	5.2	0.001	
Site	2	2.2	0.001	
Source*Site	2	1.7	0.002	
One-way analyses				
Low	1	3	0.01	
Medium	1	2.5	0.006	
High	1	3.2	0.017	

Microbes that determine changes in composition. Eighty-eight ASVs were differentially expressed between sponges from the low- and high-pollution sites (edgeR results; Fig. S4). From them, 15 ASVs belonging to Pseudomonadota, Cyanobacteriota, Campylobacterota and Bacteroidota, and unidentified ASVs were present in sponges from the low-pollution site, while 73 ASVs from the phyla Pseudomonadota, Chloroflexota, Bacteroidota, Actinomycetota, Nitrospirota, Nitrososphaerota, Planctomycetota, Acidobacteriota, and Nitrospinota were present in sponges from the high-pollution site (Fig. S4A). Differences in the microbiome between sponges from low- and medium-pollution sites were caused by the presence of five ASVs from the phylum Pseudomonadota in sponges from the low-pollution site, while the other five ASVs from the phyla Pseudomonadota, Chloroflexota, Bacteroidota, and Spirochaetota were present in sponges from the medium-pollution site (Fig. S4B). Lastly, differences in the microbiome between sponges from the high- and medium-pollution sites were due to 89 ASVs from the phyla Pseudomonadota, Bacteroidota, Actinomycetota, Nitrospirota, Calditrichaeota, Planctomycetota, Acidobacteriota, Nitrospinota, and Spirochaetota present in sponges from the high-pollution site; while seven ASVs from the phyla Pseudomonadota, Cyanobacteriota, and Spirochaetota were present in sponges from the medium-pollution site (Fig. S4C).

Ratio of common (among sponges) to opportunistic microbes

Among sponge-exclusive ASVs (absent from water samples), 65% were detected in other sponges, such as H. perlevis, H. heliophila, Haliclona sp., Tethya aurantia, Halichondria okadai, among other (Table S4). The identity of the matching sequences ranged from 92% to 100%, while the E-value ranged from 2.95E−127 to 1.73E−66, indicating robust alignments. Additionally, the 25% and 26.3% of the ASVs responsible for the differences between sponges from low- and high-pollution sites, respectively, were common to other sponge species (Table S4). The identity of the matching sequences ranged from 87% to 100%, while the E-value ranged from 2.99E−127 and 3.21E−72.

Functional analysis

A total of 6,945 KEGG ortholog (KO) genes were predicted by PICRUST2 analysis (Fig. S5), where sponge samples yielded 5,425 ± 219 genes and water samples 6,089 ± 197 genes. Among sponges, those from the low-pollution site presented 5,559 ± 170, those from the medium-pollution site presented 5,442 ± 255 and those from the high-pollution site presented 5,275 ± 154 genes. The predicted KO profile differed among the microbiomes of sponges from the different sites (PERMANOVA, F2,14 = 5.4, R2 = 0.46, P < 0.001; BETADISPER, P = 0.68) and pairwise comparisons revealed that the profile of the sponge from the high-pollution site differed from sponges of the other two sites (multilevel pairwise comparison test; PV and CTL vs. SAO P-adjusted values <0.05, PV vs. CTL p-adjusted values = 0.53; Fig. S6). Sponge microbiomes from the high-pollution site were enriched in genes associated with xenobiotic degradation and metabolism, carbon fixation, metabolism (of sphingolipids, carbohydrates, and arachidonic acid), and biosynthesis of unsaturated fatty acids and secondary metabolites (Figs. S5, S7). Conversely, sponge microbiomes from the low-polluted site were enriched with genes associated with photosynthesis, oxidative phosphorylation, metabolism (of fatty acids, glycerophospholipids, carbohydrates), and biosynthesis of peptidoglycan and lipopolysaccharide (Figs. S5, S7).

Discussion

We found that chronic water pollution was associated with significant changes in the microbiome of H. perlevis. This was evidenced by most of the evaluated indicators, especially those of taxonomic β-diversity and microbial functions, which indicated the reshaping of the microbial assemblage with the potential to acclimate sponges to environmental conditions generally considered to be adverse.

Overall, H. perlevis isolated from SAB harbors an extremely diverse microbiome of Bacteria and Archaea, different from that of seawater. The microbial assemblage of H. perlevis was dominated by Pseudomonadota, Bacteroidota, and Cyanobacteriota phyla. This microbial arrangement is characteristic of other ‘low microbial abundance’ (LMA) sponges, as reported in previous studies (Alex et al., 2013; Regueiras et al., 2017; Pankey et al., 2022).

We found that H. perlevis from SAB had a similar microbial richness along sites with contrasting pollution histories, suggesting high stability (i.e., resistance) of the microbial α-diversity along the environmental gradient, similar to other pollution-resistant sponges. The sponges Crambe crambe and Gelliodes obtusa harbor stable microbiomes, showing similar richness when exposed to nutrients and heavy metals pollution, which suggests the resistance of these sponges’ microbiome (Gantt, López-Legentil & Erwin, 2017; Baquiran & Conaco, 2018). In contrast, H. heliophila, inhabiting the polluted waters of Guanabara Bay, Brazil, exhibited a richer microbiome in comparison to counterparts in a less polluted offshore environment (Turque et al., 2010), a phenomenon the authors posit as indicative of acclimatization. With similar pollution levels to those in our study area, the difference between the findings of Turque et al. (2010) and ours may stem from their exclusive focus on archaeal representatives of the microbiome, while we assessed both the archaeal and bacterial microbes within H. perlevis.

The microbial evenness of H. perlevis from SAB changed across sites unexpectedly, with sponges from the low-pollution site exhibiting the lowest evenness while those from medium-pollution sites showed the highest evenness. Few studies have evaluated the microbial evenness of sponge microbiomes exposed to contrasting pollution conditions (Turon et al., 2019; Taylor et al., 2021). Microbial dominance, a measure related to evenness, was found to decrease in holobionts with early signs of disease because some rare microbes became more abundant and abundant microbes became less abundant (Blanquer et al., 2016). Based on changes to microbial evenness, our results point to the dysbiosis of H. perlevis at the medium and high polluted sites. However, the non-linearity between evenness and environmental disturbance assessed in our study (low, medium, and high pollution) suggests that other factors may be acting simultaneously.

Unlike with evenness, the observed differences in the microbial composition of H. perlevis points to the acclimatization of the sponge to the pollution of SAB. We predicted that both acclimatized and dysbiosis states may result in different composition shifts of the microbial assemblage. We predicted that acclimatization would decrease compositional similarity between sponge microbiomes (inter-site similarity), which was observed between high-polluted and low-pollution sites. We also predicted dysbiosis may result in decreased similarity among sponge microbiomes (lower intra-site similarity) at polluted sites, since stochastic changes in the microbiome produced by stress induces dispersion effects on the microbial assemblage composition (Zaneveld, McMinds & Vega Thurber, 2017). Conversely, we expected acclimatization to result in increased intra-site compositional similarity. We found no differences in the intra-site sponge microbiome similarity among sites. Few studies have tested the intra-site (or -treatment) similarity as a measure of stability in the sponge-microbiome association, and most of the studies where intra-site similarity was tested, it is generally done solely with the intention of validating changes among sites or treatments (inter-site effect). The temperate Australian sponge Scopalina sp. exposed to 5 days of temperature stress, showed a shift in the microbial composition (inter-site effect) without differing in the intra-treatment similarity to sponges before exposure to increased temperature (Table S4 and Fig. S5 in Taylor et al., 2021). Although, upon persistence of stress, the holobiont eventually undergoes disease and necrosis, the previously detailed changes in the microbiome of Scopalina sp. could represent an alternative state of the holobiont to acclimate to the changing environment. Previous analyses have concluded microbial stability upon finding no change in inter- or intra-site similarity (Turon et al., 2019) and instability upon finding no inter-treatment but decreased intra-treatment similarity (Lesser et al., 2016). Thus, the changes observed in H. perlevis from the SAB constitute the first record of changes in the microbial composition that suggest the establishment of new stable relationships and reconfigure itself in response to prevailing environmental conditions or stress. The compositional change in the microbiome of H. perlevis resulted from the acquisition of Nitrosopumilus, Nitrospira, Nitrospina and different unidentified ASVs belonging to Alpha- and Gammaproteobacteria (Phylum Pseudomonadota) and Chloroflexota, at the high-pollution site. While the loss of these taxa has been implicated in the deterioration of holobiont health (Botté et al., 2023), their acquisition may enhance holobiont homeostasis (Zhang et al., 2019; Marangon et al., 2021). Representatives of Nitrosopumilus, Nitrospira and Nitrospina are recognized for their roles in the nitrogen cycle. By assimilating nitrogenous waste products derived from the host or eutrophic waters, these microbial additions may benefit H. perlevis in challenging environments such as SAB.

Hymeniacidon perlevis exposed to high and chronic pollution at SAB harbored the same ratio of common to opportunistic microbes as sponges from the low-pollution site. This indicates that changes in the microbial assemblage of sponges from the low to the high-pollution site was not due to the acquisition of opportunistic and non-endemic microbes, but to shifts in the relative abundances of existing microbial members and recruitment of new microbes with established sponge associations. In agreement with the β-diversity indicators, the paucity of non-endemic microbes points to the acclimatization of H. perlevis to high-pollution conditions. To the best of our knowledge, this ratio has been evaluated in a few sponge holobionts and this is the first report of visually healthy, non-necrotic H. perlevis individuals. Necrosis is thought to arise following extreme dysbiosis and is associated with an 80% reduction in the ratio of common to opportunistic microbes (Egan & Gardiner, 2016; Pita et al., 2018; Webster et al., 2010; Simister et al., 2012a). Thus, we expected that an increase in the state of dysbiosis, towards disease and necrosis, will be accompanied by a decrease in the ratio of common to opportunistic microbes. While the current work does not find evidence of opportunistic microbes or dysbiosis, further studies are needed relating changes in the ratio of common to opportunistic microbes to different health states of holobionts under different disturbance scenarios.

Consistent with the taxonomic β-diversity results, the functional analysis using PICRUSt2 evidenced a change in the predicted microbial functional profile of sponges exposed to high and chronic pollution at SAB. In contrast to expectations under dysbiosis, functions related to virulence, cell motility or chemotaxis were not differentially expressed across sites. Instead, our results suggest that the introduction of new microbial functions may benefit sponges in sites with diverse pollutants, through the degradation of diverse xenobiotic compounds and exploitation of greater trophic supply (dissolved and particulate compounds), resulting in acclimatization of H. perlevis at the polluted site. Additionally, some differentially abundant ASVs in the sponges from the high-pollution site were affiliated with nitrifying taxa common to ammonia-rich environments, including wastewater and sponges (Off, Alawi & Spieck, 2010; Reveillaud et al., 2014; Subina, Thorat & Gonsalves, 2018). For example, members of Nitrosopumilaceae, specifically two ASVs affiliated with Nitrosopumilus, two Nitrospina ASVs, and three Nitrospira ASVs were enriched in sponges from the high-pollution site. Nitrosopumilus is a group of Archaea with ammonia oxidizing functions. Nitrospina and Nitrospira, on the other hand, are nitrite-oxidizing bacteria, which often occur in close association with ammonia-oxidizing bacteria (Daims & Wagner, 2018). Altogether, nitrifying bacteria and archaea potentially contribute to ammonia detoxification and nitrogen cycling (Feng et al., 2016).

In summary, the microbiome shift among sites with different pollution histories at SAB suggests a mechanism by which the holobiont may acclimatize to a wide range of polluted conditions. Indeed, long-term monitoring of H. perlevis at these sites (since 2012) indicates a thriving population with no mass mortality events and high substrate cover (up to 30% of the substrate) in highly polluted areas (Gastaldi et al., 2016, 2017, 2020). Furthermore, the acclimatization of H. perlevis at the high-pollution site may be favored by the macrotidal nature of the bay, where a great water exchange (twice a day) dilutes the pollutants and nutrients preventing the development of the dysbiosis process in the sponges inhabiting that impacted site. While our findings suggest the potential adaptability of H. perlevis to polluted environments through microbiome rearrangement, limitations including geographic and temporal constraints and sample size, underscore the necessity for further testing. The dynamic nature of environmental conditions and anthropogenic stressors necessitates long-term monitoring efforts to capture temporal changes in sponge microbiomes; sampling from a larger geographic scale would also provide a more comprehensive understanding of sponge-microbe interactions in the face of anthropogenic stressors. Furthermore, future research should be conducted to experimentally test whether a dysbiosis scenario could arise with the conditions compounded by growing anthropogenic stressors.

Supplemental Information

Supplemental Information 1 Evolutionary relationships of Hymeniacidon perlevis.

Relationships were recovered using CO1 (A), 18S (B), and ITS2 (C) sequences. Phylogenetic reconstructions recovered through the Maximum Likelihood (ML) and Bayesian methods. Numbers at nodes are Bootstrap support values (ten and thirty thousand, respectively). Hymeniacidon sinapium is presented as H. perlevis following Turner 2020 (see Table S2 for sequences references).

Supplemental Information 2 nMDS ordination plot of Hymeniacidon perlevis and water microbial assemblages based on Jaccard dissimilarity.

Ellipses indicate a confidence level of 80%. Ordination stress is 0.1. Low, medium, and high refer to sites with different pollution levels. The description of pollution levels can be found in Table S1.

Supplemental Information 3 Within-site microbial compositional dissimilarity between Hymeniacidon perlevis and water.

Low, medium, and high refer to sites with low, medium and high pollution. The description of pollution levels can be found in Table S1.

Supplemental Information 4 Differentially expressed ASVs among Hymeniacidon perlevis from sites with different levels of pollution.

A) Microbes of sponges from sites with low- and high-pollution, B) with low- and medium-pollution, and C) with high- and medium-pollution.

Supplemental Information 5 Heatmap showing differentially enriched KEGG pathways among Hymeniacidon perlevis and water samples.

Low, medium, and high represent sites with different levels of pollution (see Table S1). Numbers in Site horizontal color bar indicate the number of samples.

Supplemental Information 6 MDS ordination plot based on Jaccard dissimilarities of KO genes of Hymeniacidon perlevis.

Low, medium, and high represent sites with different levels of pollution (see Table S1). Stress level=0.065.

Supplemental Information 7 Bar plot of the differentially enriched KEGG pathways in Hymeniacidon perlevis from the sites with contrasting pollution histories.

Low and high represent sites with contrasting levels of pollution (see Table S1).

Supplemental Information 8 Concentration of nutrients and heavy metals pollutants previously reported for the studied sites.

Canada Government guidelines reference values are included for reference. ND: non-detected concentration; NA: not available data.

Supplemental Information 9 Summary table of the Hymeniacidon perlevis sequences used for phylogenetic analyses.

* indicate haplotype sequences used in the analyses. CTL, PV, and SAO indicate sites with low, medium, and high pollution level. The description of pollution levels can be found in Table S1.

Supplemental Information 10 Microbial phyla present in water and Hymeniacidon perlevis samples.

ASVs abundance is reported as raw counts, percentage of grouped samples, and percentage of abundance at each site. Low, medium, and high refer to sites with different pollution levels. The description of pollution levels can be found in Table S1.

Supplemental Information 11 BLASTN results reporting the five first matches.

in the Condition column, “sponge exclusive” refers to matches of ASVs that were differentially expressed in sponge samples compared to water samples (analysis 1); in contrast, “low exclusive” and “high exclusive” refer to matches of ASVs that were differentially expressed in sponge samples from low- and high-pollution sites, respectively (analysis 2).

Supplemental Information 12 Supplemental Article S1.

Supplementary information on ITS2 amplification and sequencing, 18S, COI, 16S amplifications and pipeline, and the statistical approach used in this study.

We wish to thank the reviewers whose suggestions have made a significant improvement in the manuscript. We are also grateful to Eliana Gastaldi, Patricio Pereyra, Juan Saad, and Gonzalo Landete for their help during samplings.

Additional Information and Declarations

Competing Interests

Author Contributions

DNA Deposition

Data Availability

The authors declare that they have no competing interests.

Marianela Gastaldi conceived and designed the experiments, performed the experiments, analyzed the data, prepared figures and/or tables, authored or reviewed drafts of the article, and approved the final draft.

M. Sabrina Pankey analyzed the data, authored or reviewed drafts of the article, and approved the final draft.

Guillermo Svendsen performed the experiments, analyzed the data, authored or reviewed drafts of the article, and approved the final draft.

Alonso Medina analyzed the data, authored or reviewed drafts of the article, and approved the final draft.

Fausto Firstater performed the experiments, authored or reviewed drafts of the article, and approved the final draft.

Maite Narvarte performed the experiments, authored or reviewed drafts of the article, and approved the final draft.

Mariana Lozada analyzed the data, authored or reviewed drafts of the article, and approved the final draft.

Michael Lesser performed the experiments, authored or reviewed drafts of the article, and approved the final draft.

The following information was supplied regarding the deposition of DNA sequences:

The sequences in this study are available in GenBank under: MZ292042, MZ297336–MZ297354 (COI sequences); MZ298268–MZ298287 (18S sequences); and MZ435928–MZ435947 (ITS2 sequences).

The microbiome 16S rRNA libraries are available in NCBI Short Read Archive: PRJNA734169.

The following information was supplied regarding data availability:

The microbiome 16S rRNA libraries are available in NCBI Short Read Archive: PRJNA734169.

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
