# Peer review of "Holobiont dysbiosis or acclimatation? Shift in the microbial taxonomic diversity and functional composition of a cosmopolitan sponge subjected to chronic pollution in a Patagonian bay"

_PeerJ, doi:10.7717/peerj.17707_

## Round 0.1 · original submission · Major Revisions

Please provide a detailed point-by-point rebuttal letter to each of the reviewers' comments, along with your revised manuscript. The provided comments are very helpful and accurate for improving your manuscript.

·

Basic reporting

This study reports the detailed microbiome composition of H.perlevis, which is inclusive of bacteria and archaea from different sites. The authors have chosen low to high-pollution sites, in Antonio Bay, Patagonia, Argentina. The study is of importance. The authors have raised the question of Dysbiosis and Acclimatization and in the conclusion pointed towards the phenomenon of acclimatization and not dysbiosis

Experimental design

As the Anthropogenic pressures are increasing and the pressures that Sponges are facing are becoming a concern, this study focuses on the same. Sponges being from the benthic community that acts as water cleaners, it is of prime importance to understand what kind of pressure they feel and under such pressure, if their water cleaning ability is disabled. There is a cost attached to the survival of the sponge.

The word Pollution is used in this manuscript without assessing the kind of pollution. This disconnects the study and its importance which could have of utmost interest.

Validity of the findings

The findings are reported as of course, microbiome data, which the Sequencing data gives and there is no question about that.

Major Experiments which can be performed

1. At Each site, pollution data must be collected, kind of pollution, the concentration of heavy metals, antibiotics, etc. Each water from where the sponges are collected must be sampled for pollution.

2. This must be a dynamic time-dependent study, which must be performed after every month, till say for a year, to understand whether the Sponge is still retaining the water cleaning property along with its change in the gut flora.

3. The pictures of H.pervelis shown in Figure 2, are they all on same scale ( 5 cm)? There is a massive visual difference in Figure 2B compared to A or C

4. Dysbiosis and acclimatization are sometimes difficult to dissect. It can start with dysbiosis and later lead to acclimatization. As these experiments are all done on sites and nothing is in lab-bound controlled conditions, certain features are difficult to conclude.

Reviewer 2 ·

Basic reporting

This is an interesting study that is well thought out and has clear goals and conclusions. The research is novel and contributes to our understanding of sponge ecology in an ocean facing increasing anthropogenic influence. However, the grammar throughout could be improved, as there are times where it is slightly difficult to follow the author's arguments.

Additionally, while all figures included are relevant to the article, there are a few that may be more appropriately included in a supplemental file.

The largest area for improvement is the methods. There is much superfluous information included in the methods that would be more appropriately placed in either the introduction or discussion sections.

Experimental design

The experimental design is appropriate and well laid out.

The research question is well-defined and meaningful, and clearly fills a gap in our knowledge about how sponges, a dominant member of the benthos, are affected by pollution.

As stated previously, while the methods are detailed and provide sufficient information, there is also a lot of information included in the methods that would be more appropriate in either the intro or discussion.

Validity of the findings

The findings seem to be valid and there is sufficient replication.

Underlying genetic data have been supplied. However, while great detail was given on how data were analyzed using R, I was unable to find a link to the raw data or code used to run analyses.

Conclusions are well-stated for the most part, though greater emphasis could be placed on connecting findings from the current study to findings from previous studies (see attached PDF for more detailed feedback).

However, the findings overall are understandable, and represent what I believe to be an accurate interpretation of the data.

Annotated reviews are not available for download in order to protect the identity of reviewers who chose to remain anonymous.

Reviewer 3 ·

Basic reporting

This manuscript evaluates if Hymeniacidon perlevis enters a dysbiosis or acclimatization state as a response to water pollution. Authors hypothesize that the successful colonization of heavily impacted sites by H. perlevis is related to reshaping its microbiome, which facilitates holobiont acclimatization in this impacted environment. To test their hypothesis, authors assessed shifts in microbial taxonomic composition and functional profiles in sponges collected at three sites with historically different water pollution levels. Authors’ predictions to define if H. perlevis enters dysbiosis or acclimatization are based on the following definitions (I try my best to collect and synthesize all definitions that are dispersed throughout the text. I suggest authors include a table or scheme that will allow readers to find all the definitions).

1. Resistance and resilience = healthy sponge and sponge-microbes association remains stable under environmental disturbance.
2. Dysbiosis or acclimatization = potential signs of disease on the sponge (based on previous studies), and changes in richness and composition of microbial assemblages.

2.1) Dysbiosis = an increase in microbiome evenness, increase in similarity between the holobiont and the environment microbial communities, decrease in microbiome similarity intrasite (= within sponge samples in the same site or pollution treatment) in relation to intersite (= between sponge samples in the same site or pollution treatment).

2.2) Acclimatization = a decrease in microbiome evenness, dissimilarity between host and environmental microbial communities, increase in microbiome similarity intrasite (= within sponge samples in the same site or pollution treatment) in relation to intersite (= between sponge samples in the same site or pollution treatment).

To distinguish between holobiont outcomes (i.e., dysbiosis or acclimatization), authors analyzed a set of indicators of the microbiome including 1) taxonomic alpha-diversity (ASV richness, and ASV evenness), 2) taxonomic beta-diversity, the contribution of specific taxa to the observed differences between sources and among sites, 3) the ratio of common to opportunistic microbes, and 4) microbial functions (metabolic capability). Results suggested H. perlevis acclimates to high levels of pollution.

This study is relevant to understanding how sponges acclimate and successfully colonize sites with high anthropogenic impact while resisting dysbiosis.


Main Issues

1. The manuscript’s order of ideas jumps back and forth to define what characteristics in the holobiont suggest dysbiosis or acclimatization. Organizing this information is critical to guiding readers through your hypotheses and understanding the results. As indicated in the previous section, please include a table or diagram/scheme with all the information concisely.

2. The English language should be improved. Several sentences lack proper grammar and are cluttered with unnecessary prepositions and repeated words, which makes it harder to understand the authors’ statements, hypotheses, ideas, and overall interpretation of results.

3. Experimental Design—ratio of common and opportunistic microbes—It is unclear why only the first five matching sequences were included in the analysis. Also, it is unclear why the search was not restricted by the type of microbial abundance, HMA or LMA, or limited to a specific geographic region, as both factors can contribute to increasing bias.


Formatting and Language

The English language needs improvement; the grammar and style switch drastically across the document.

Line 57 – “is projected to be from 50-120% for the period 2030-2060”. Clarify, "is projected to change from xx to xx for".

Line 85- “In both dysbiotic and acclimatized outcomes, microbial assemblages were found to change in terms of richness and composition” Delete- “were found to”.

Lines 121-125- First state your objective, then the hypothesis. Also, reword the objective “the objective of this study was to distinguish between a state of acclimatization and dysbiosis of the sponge holobiont through a comprehensive study of microbiome indicators.” Suggested change –use similar wording for the objective as you have in the abstract – “assess whether the shift …..etc..”

Line 197 – Mention why you use ASV and spell the full term at its first mention.

Line 223-225- “holobiont acclimatization is associated with changes in the microbial compositional that increase the dissimilarity between host and environmental communities”. Declutter this sentence from unnecessary words- holobiont acclimatization is associated with an increase in microbial composition that differs from the environment.

Lines 225-229 – Needs rewording.

Lines 230-231 – “we predict that: a) high compositional similarity between sponge and water microbiomes and/or b) an increase in their similarities towards the more polluted site signals dysbiosis,”. Reword for “we predict that dysbiosis is characterized by a) high compositional similarity between sponge and water microbiomes and/or b) an increase in their similarities towards the more polluted site signal”

Lines 233-238- convoluted sentences – declutter and reword.

Lines 311- 320—Rephrase: First, describe the similarities between the sponge and the water and then the differences.

Line 377- repetition of “sponges” in the same sentence. Please reword.

Lines 395-398- Unnecessary clarification; instead, explain what your findings mean in lines 393-395.

Line 413- If Turque et al. (2010) only considered the archaeal representative, please include the word “only.”

Lines 419-420: “In other studies, indicators related to evenness were used, which gives us some insight on microbial evenness." This sentence is vague—it does not provide any relevant information and does not connect well with the sentences before and after. Please state clearly how evenness indicators from other studies help you understand your results.

Lines 421-423, “Microbial dominance was found to increase in holobionts with early signs of disease as a result of the increase in abundance of rare microbes and the decrease of abundant microbes, was found (Blanquer et al., 2016).” This sentence does not connect with the previous sentence. Also, the verb is placed alone after a comma. This sentence requires both grammar check and clarification-style revision.

Lines 437- 439, the sentence/statement is incomplete. Please finish the statement and explain what it means in the context of your study.

Lines 442-443- The word increased is repeated twice; you could change the increased temperature for temperature stress. Also, please add the temperature values in parentheses for the treatment and control.

Line 445- Change “All in all” for a better connector word.

Lines 447-448- The statements in both paragraphs seem connected. If yes, please connect the sentences.

Lines 463- 467- Mention if you observe any necrotic state in H. perlevis when comparing with R. odorabile. Is this comparison valid across a species experiencing necrosis and one that did not? Expand the discussion here, including host health (healthy and necrotic) vs. change in the microbiome.

Move Lines 482- 498 to line 470, after the word pollution. It is better to state your findings (what is relevant) and then contrast the patterns you found with other studies. The order of these paragraphs does not connect your findings well with previous research.

Line 513-514- “remains to be tested”- please explain in which context it remains to be tested. Do you mean experimentally in the lab? Please expand on your study limitations and why it needs to be tested.

Figures

Figure 1—Mark each panel as a,b, or c and reference it in the figure caption accordingly. Add the pollution level beside the site name to the legend in the second figure. Simplify figure caption language and be consistent when using acronyms and full names (e.g., San Antonio Bay has an acronym, but San Antonio Oeste does not). Ensure all maps have a compass rose and a scale bar in km.

Figure 2—The description of the figure is not clear. Please reword as follows: “collected at sites with different levels of pollution: A) low, B) medium, and C high. The description of pollution levels can be found in Table xx. Add a scale bar to all panels in the figure, or explain in the figure description that the scale bar in A is the same for B and C.

Figure 3—Reword the figure description and avoid repeating low, medium, and high. The Y-axis on panels A and B should follow the same order and range from lower to higher pollution. For example, if you place sponge-low first, the logic is that the next one will be sponge-medium and then sponge-high; do the same with water samples and keep it consistent in any supplemental figure or other figures in the document.

Figure 4- Reword the figure description, and avoid repeating low, medium, and high. On the x-axis, organize the names in a logical order from low to high; for example, sponge-low first, the logic is that the next one will be sponge-medium, and then sponge-high; do the same with water samples and keep it consistently in any supplemental figure or other figures in the document.

Figure 5—The description of the figure is convoluted. Please revise English grammar and style. Also, explain what the left color bar on each panel of the figure means (relative abundance?) and what each dot on the left means (presence and absence?). Why is this figure represented as the log 2-fold change? Please explain.

Figure 6- The heatmap needs to be reorganized; the second row of sites should be organized from low to high pollution (also add the number of samples per site on the figure description, as it seems the design is unbalanced, but it is supposed to be five samples per site). Also, reorganize the path class on the left (left color bar) following the same order as in the legend-path class; that way, it is easier to visualize patterns per each factor (source and pollution).

Figure 7—Include in the figure description what proportion of the variance of the scaled data was accounted for by the MDS procedure (axis values) and reliability (e.g., stress value).

Figure 8—Color code or group each type of pathway on the figure, for example, lipid metabolism in one group, carbohydrate metabolism in another group, etc.


References

Please revise- species names should always be in italics.

Experimental design

The research question is very interesting and meaningful; however, the way in which the authors proposed to test dysbiosis or acclimatization is scattered in the text. The authors should make an experimental design scheme/flow diagram with all the variables measured and how they are combined to test either dysbiosis or acclimatization. This scheme will help readers better understand the design and the patterns of microbiome indicators that authors use to test their hypotheses.

Explain why n=5 samples per site were collected. Also, explain why the authors used only one site per type of pollution and not more (for example, three sites per pollution level). Site replication per pollution level would strengthen the study.

Explain in the text the general steps to obtain amplicon sequence variants (ASV) for the analysis and why this method is appropriate.

The ratio of common and opportunistic microbes - It is unclear why only the first five matching sequences were included in the analysis. Please explain. Also, it is unclear why the search was not restricted to the type of microbial abundance, HMA or LMA, or a specific geographic region, as both factors can contribute to increasing bias.

Validity of the findings

The study results are novel and relevant to understanding how sponges acclimate and successfully colonize sites with high anthropogenic impact. However, the number of sites with different pollution levels is low (only one site per pollution level). Also, the number of samples (n=5) per site could be higher (n=10). The authors must evaluate the limitations of this study in the discussion.

It is important to mention if H. perlevis is an HMA or LMA sponge, as functional gene content, pumping rate, and exchange of carbon and nitrogen compounds differ regarding the HMA-LMA dichotomy. The study's findings are relevant to the type of microbial abundance this species holds. Please mention this in the introduction and the discussion. When comparing with other studies, you are comparing both HMA and LMA sponges, and the differences between your research and previous ones may be attributed to the dichotomy.

Explain why an ASV approach may be more relevant than an OUT approach and why you utilized an ASV approach.

Line 324—results—Taxonomic alpha-diversity—Explain what ASV richness variation means for the interaction term and in the context of your study. Mentioning that the interaction terms differ does not give any information. Instead, explain what the interaction means regarding sample sources and sites.

---

## Round 0.2 · accepted · Accept

Thank you for your revised version, which can now be accepted for publication.

·

Basic reporting

The Authors have reported the ability of Sponges to clear water from its pollutants.

Experimental design

As suggested, all the experiments are in the pipeline for a more exhaustive study. Once done, that can become a paper of importance. However, other comments have been addressed

Validity of the findings

The findings are very clean and of environmental interest. Though I was very keen on the other experiments, I can also appreciate the limitations. However, it is important to note that the authors are doing an exhaustive study.

Additional comments

Dear Editor-in-Chief
The manuscript can now be accepted.
Thanks for this opportunity
Regards
Dipshikha